# TRELLIS: Learning to Compress Key-Value Memory in Attention Models

**Mahdi Karami, Ali Behrouz, Praneeth Kacham, Vahab Mirrokni**
Google Research
{mahdika,alibehrouz,pkacham,mirrokni}@google.com

## Abstract

Transformers, while powerful, suffer from quadratic computational complexity and the ever-growing Key-Value (KV) cache of the attention mechanism. This paper introduces Trellis, a novel Transformer architecture with bounded memory that learns how to compress its key-value memory dynamically at test time. Trellis replaces the standard KV cache with a fixed-size memory and train a two-pass recurrent compression mechanism to store new keys and values into memory. To achieve this, it leverages an online gradient descent procedure with a forget gate, enabling the compressed memory to be updated recursively while learning to retain important contextual information from incoming tokens at test time. Extensive experiments on language modeling, common-sense reasoning, recall-intensive tasks, and time series show that the proposed architecture outperforms strong baselines. Notably, its performance gains increase as the sequence length grows, highlighting its potential for long-context applications.

## 1 Introduction

Transformers (Vaswani et al., 2017) has established itself as the *de facto* architecture for sequence modeling in modern deep learning, achieving significant advances across diverse areas, including language modeling (Devlin et al., 2018; Radford et al., 2018), computer vision (Arnab et al., 2021; Dosovitskiy et al., 2020), and graph learning and generation (Dwivedi & Bresson, 2020; Karami, 2024; Yun et al., 2019). Their success stems from the attention mechanism, which allows models to dynamically attend to relevant parts of an input sequence while enabling parallel computation. This enables the capture of long-range dependencies and in-context learning. However, the quadratic time and space complexity of attention with respect to sequence length restricts its scalability in long sequence modeling. Furthermore, the requirement for an *unbounded cache* leads to inefficient memory management, particularly in resource-constrained environments.

These limitations have driven the exploration of alternative architectures that aim to retain the representational power of Transformers while addressing their computational and memory complexity. One key strategy involves *sparsifying* the dense attention matrix through various techniques, including: blockwise attention (Parmar et al., 2018; Qiu et al., 2019); using strided or sliding window attention patterns (Beltagy et al., 2020; Child et al., 2019; Zaheer et al., 2020); or clustering/sorting tokens (Kitaev et al., 2020; Roy et al., 2020; Tay et al., 2020). Another approach involves *low-rank approximations* of the self-attention matrix, leveraging the insight that it often exhibits low-rank properties (Wang et al., 2020). A different paradigm employs the kernel trick, replacing the Softmax operation with a dot product of feature maps, resulting in a family of linear attentions (Choromanski et al., 2020; Katharopoulos et al., 2020; Peng et al., 2021b). While these methods substantially reduce computational costs, they may sacrifice expressiveness and performance, often requiring hybrid approaches that combine them with dense attention layers (Fu et al., 2023; Mehta et al., 2022). Additionally, global convolutions (Li et al., 2022; Poli et al., 2023; Romero et al., 2021) and their input-dependent variants (Karami & Ghodsi, 2024) have been explored as alternative sequence modeling techniques.

Recently, Recurrent Neural Network (RNN) architectures have re-emerged as promising attention-free solutions for sequence modeling. These models leverage the parallelization capabilities of their linear recurrence, building upon earlier linear time-invariant models (Fu et al., 2023; Mehta et al., 2022; Wang et al., 2022) and extending them to more expressive input-dependent gated RNN designs with linear memory (De et al., 2024; Gu & Dao, 2023; Orvieto et al., 2023; Yang et al., 2024b) that demonstrate improved in-context learning while retaining computational advantages. However, their effective memory is limited, affecting their ability to efficiently compress and summarize information over very long contexts within their fixed-size hidden states.

**Our approach and main contributions** In this paper, we present Trellis, a new Transformer architecture with bounded memory that learns how to compress its key-value memory at test time. To address the linearly-growing memory of global attention, Trellis replaces the KV-cache (i.e., the pair of $K$ and $V$ matrices) with a bounded memory with $m$ slots and train a meta in-context model to learn how to store new keys and values into memory. For better management of memory, Trellis uses a forgetting mechanism that learns how to selectively forget unnecessary past information in its compressed KV-cache.

We perform an extensive set of experimental evaluations on various tasks including language modeling, commonsense reasoning, recall-intensive, needle-in-haystack, and time series data. We observe that Trellis outperforms state-of-the-art baselines, including Transformer++, and modern linear recurrent neural networks in downstream tasks. Furthermore, Trellis scales better than baselines when increasing the context length, showing promising results for long-context tasks.

## 2    Related Work and Background

For an input sequence $\mathcal{X} = [x_1, \ldots, x_T]$, where $x_t \in \mathbb{R}^d$, the *causal Softmax attention* mechanism generates output tokens $y_t \in \mathbb{R}^d$, by attending to past tokens (historic context):

$$y_t = \mathcal{V}_t \, \text{Softmax}(\mathcal{K}_t^\top \, q_t) \,. \tag{1}$$

Here, the *query*, *key*, and *value* vectors are computed by linear projections of the input: $q_t = \mathbf{W}_q \, x_t$, $k_t = \mathbf{W}_k \, x_t$, $v_t = \mathbf{W}_v \, x_t$, where $\mathbf{W}_q, \mathbf{W}_k, \mathbf{W}_v \in \mathbb{R}^{d \times d}$ are learnable weight matrices. The key-value memory, represented by the caches $\mathcal{K}_t \in \mathbb{R}^{d \times t}$ and $\mathcal{V}_t \in \mathbb{R}^{d \times t}$, stacks the key and value vectors of each new token, leading to unbounded caches with linearly-growing size. The retrieval of relevant information from this key-value cache can be rewritten as a weighted sum:

$$y_t = \mathcal{V}_t \, a_t, \ \ \text{where } a_t = \text{Softmax}(\mathcal{K}_t^\top \, q_t) \in \mathbb{R}^t.$$

Here, the vector $a_t \in \mathbb{R}^t$ is the collection of the attention scores between $t$-th token and its historic context. Hence, the attention in equation (1) can be seen as a non-linear query from an unbounded memory.

The key-value cache size growth poses a significant memory bottleneck during inference, especially for long sequences. Additionally, each retrieval operation scales linearly with sequence length, resulting in an overall quadratic computational complexity $\mathcal{O}(T^2)$ for generating a full sequence of length $T$.

To address the computational and memory bottleneck of Softmax attention, various alternatives have been proposed (Tay et al., 2022). A well-established approach involves employing the kernel trick to replace the Softmax with a dot product of feature maps, $\phi(q_t)$, $\phi(k_t)$ commonly known as Linear Attention LA (Katharopoulos et al., 2020).

To maintain bounded computational and memory requirements in attention mechanisms, an alternative strategy is to explicitly use a fixed-size key-value cache. In this approach, the memory matrices **K** and **V** are constrained to a length $m$ where $m \ll T$. A straightforward implementation of this strategy involves limiting the attention window to the most recent $m$ tokens by maintaining a first-in-first-out (FIFO) queue, often referred to as Sliding Window

Attention (SWA). While SWA achieves linear computational complexity, it suffers from a limited receptive field, restricting the model's ability to capture long-range dependencies and leading to a poor recall-memory trade-off (Arora et al., 2024). Moreover, the observation, supported by many research works that key-value matrices in attention mechanisms often exhibit low-rank and sparse structures (Chen et al., 2021; Singhania et al., 2024; Wang et al., 2020) motivates the design of efficient sequence mixing layers that induce these properties. Therefore, these layers aim to compress the context, storing the important information while discarding or forgetting redundancies, rather than naively truncating memory.

In light of this insight, the Attention-with-Bounded-Memory Control (ABC) mechanism (Peng et al., 2021a) introduces a method to compress and dynamically update a fixed-size memory. This is achieved using a pair of linear recurrences to update the key and value matrices:

$$\mathbf{K}_t = \mathbf{K}_{t-1} + \boldsymbol{\alpha}_t \, \boldsymbol{k}_t^\top \in \mathbb{R}^{m \times d},$$
$$\mathbf{V}_t = \mathbf{V}_{t-1} + \boldsymbol{\alpha}_t \, \boldsymbol{v}_t^\top \in \mathbb{R}^{m \times d},$$
$$\boldsymbol{y}_t = \mathbf{V}_t^\top \, \text{Softmax}(\mathbf{K}_t \boldsymbol{q}_t) \in \mathbb{R}^d \tag{2}$$

Here, $\boldsymbol{\alpha}_t := \text{Softmax}(\mathbf{W}_{\boldsymbol{\alpha}} \boldsymbol{x}_t) \in (0,1)^m$ controls the update distribution across the memory slots. Each component, $\alpha_{t,j}$, can be interpreted as the writing intensity determining the $t$-th token contribution to the $j$-th memory slot.

The ABC update rule can be decomposed into a cascade of two LAs and presented in the following two-pass process:

$$\{\hat{\boldsymbol{y}}_t\}_{t=1}^T = \text{LA}(\{\boldsymbol{q}_t, \boldsymbol{k}_t, \boldsymbol{\alpha}_t\}_{t=1}^T), \; \hat{\boldsymbol{y}}_t \in \mathbb{R}^m,$$
$$\{\boldsymbol{y}_t\}_{t=1}^T = \text{LA}(\{f(\hat{\boldsymbol{y}}_t), \boldsymbol{\alpha}_t, \boldsymbol{v}_t\}_{t=1}^T), \; \boldsymbol{y}_t \in \mathbb{R}^d, \tag{3}$$

where the intermediate activation function: $f(\cdot) = \text{Softmax}(\cdot)$. Building upon this foundation, Gated Slot Attention (GSA) (Zhang et al., 2024) introduced a gated version of ABC, enhancing the two-pass process with a forget-gate mechanism introduced in GLA (Yang et al., 2024b). This mechanism allows the model to forget irrelevant information, resulting in better memory management and improved performance. Importantly, unlike many linear attention variants, both ABC and GSA do not rely on kernel approximations and retain the Softmax non-linearity. Moreover, they can take advantage of the chunkwise matrix form (Hua et al., 2022; Kacham et al., 2024; Yang et al., 2024b), enabling parallel hardware-efficient implementations on tensor cores.

The simple linear additive (Hebbian-like) nature of the recurrence in ABC and GSA, however, limits the memory capacity and effective memory management in long-context retrieval tasks (Behrouz et al., 2025b; Schlag et al., 2021). Specifically, the additive Hebbian update rule in ABC and GSA endlessly accumulates new tokens into the fixed memory space without an explicit mechanism to replace past value with new information in the memory, leading to an *overcapacity regime* (Schlag et al., 2021). Moreover, their state-independent linear outer product modification of memory lacks dynamic interaction between the memory content and incoming keys, disabling it to selectively discard irrelevant or redundant information.

To address these limitations, this paper introduces a new two-pass *non-linear recurrence*. Leveraging techniques from meta-learning and test-time memorization (Behrouz et al., 2024; Sun et al., 2024a), our approach is designed to dynamically compress new keys and values into the memory while minimizing information loss. For clarity, the subsequent explanation and its notations focus on key ($\boldsymbol{k}_t$) cache compression in the first pass, noting that the same principles apply to the value cache in the second pass.

## 3 Method

We define the compression model as a *regression layer* that projects a key token into a latent space. Specifically, for each token embedding $k_t$ and its corresponding latent representation $\alpha_t$, the compression layer aims to reconstruct a target vector such that $\alpha_t \approx \hat{\alpha}_t = \phi(\mathbf{M}_t k_t)$.

To minimize the reconstruction error and the compression loss, we formulate the learning objective as an $\ell_2$ optimization problem:

$$\mathcal{L}_t = \|\phi(\mathbf{M}_t k_t) - \alpha_t\|^2, \quad \mathbf{M}_t \in \mathbb{R}^{m \times d}, \; k_t \in \mathbb{R}^d, \; \alpha_t \in \mathbb{R}^m \tag{4}$$

We model the latent representation $\alpha_t$ using an encoder network, which is implemented as a linear projection of the input: $\alpha_t = \mathbf{W}_\alpha x_t$ with projection weight matrix $\mathbf{W}_\alpha \in \mathbb{R}^{m \times d_x}$. Our approach follows the framework of *Fast Weight Programmers (FWPs)* (Schlag et al., 2021; Schmidhuber, 1992), where the internal memory of the compression layer (*a.k.a.* state in the context of RNNs), $\mathbf{M}_t$, serves as "fast weights", which are dynamically updated based on streaming input data. Hence, each sequence serves as a training dataset for the learning in this *inner loop*. To efficiently adapt to new tokens, we design an internal learning procedure that continuously updates $\mathbf{M}_t$, allowing it to store in-context information.

In this framework, the outer network, also referred to as the "slow" network, consists of the projection layer weights and the rest of the model parameters, jointly denoted as $\mathcal{W}$. These parameters are trained in the outer loop, which follows standard deep neural network optimization process, minimizing the end-to-end loss averaged over the training dataset to learn generalizable patterns from the training set. This learning process subsequently enables fast adaptation within the inner loop (Schlag et al., 2021). Importantly, the weights of the "slow" network remain frozen during the internal state updates of the inner loop (involving the "fast" weights). This overall procedure constitutes a bi-level optimization strategy (Chen et al., 2022; Liu et al., 2022) commonly used in meta-learning (also referred to as learning to learn) (Andrychowicz et al., 2016; Bengio et al., 1990; Schmidhuber, 1992; Thrun & Pratt, 1998).

Given the sequential nature of the data, we approach this problem as an *online optimization problem* and update the internal memory using one gradient descent step per token:

$$\mathbf{M}_{t+1} = \mathbf{M}_t - \gamma_t \nabla_{\mathbf{M}} \mathcal{L}(\mathbf{M}_t, v_t, \alpha_t) \tag{5}$$

This update rule generates the sequence of states $\{\mathbf{M}_t\}_{t=1}^T$, where each state $\mathbf{M}_t$ is a nonlinear recurrent function of the previous state and the input token. Consequently, the memory update follows a causal nonlinear recurrence, ensuring that information is continually integrated into the memory.

By the chain rule, the gradient of the loss with respect to $\mathbf{M}$ can be computed:

$$\nabla_{\mathbf{M}} \mathcal{L}_t = \mathbf{G}_t(\mathbf{M}_{t-1}, \alpha_t, v_t) = 2 \left( \mathbf{J}_\phi \left( \phi(\mathbf{M}_{t-1} k_t) - \alpha_t \right) \right) k_t^\top \tag{6}$$

where $\mathbf{J}_\phi$ is the Jacobian of $\phi(\cdot)$. In practice, terms involving Jacobian products (such as $u_t^\top := \mathbf{J}_\phi \left( \phi(\mathbf{M}_{t-1} k_t) - \alpha_t \right)$ in the expression above) are computed efficiently using the vector-Jacobian product (vjp) method available in modern machine learning frameworks. This avoids explicitly forming the full Jacobian matrix and leverages efficient automatic differentiation.

### 3.1 State Decay

While the proposed compression layer can address the quadratic time and space complexity of Transformers by learning how to effectively compress key-value pairs $(k_t, v_t)$ into fixed-size memory states, these memories can still overfit to early tokens of the sequence or overflow as information accumulates. To mitigate these issues, we introduce $\ell_2$ regularization on the memory states—analogous to weight decay in standard neural network training—applied within the inner loop:

$$\mathcal{L}_t = \|\phi(\mathbf{M}_t v_t) - \alpha_t\|^2 + \frac{\lambda_t}{2} \|\mathbf{M}_t\|^2, \tag{7}$$

This regularized objective results in a gradient descent recurrent update with *state decay*:

$$\mathbf{M}_t = (1 - \lambda_t)\mathbf{M}_{t-1} - \gamma_t \nabla_{\mathbf{M}}\mathcal{L}(\mathbf{M}_{t-1}, v_t, \boldsymbol{\alpha}_t) = \beta_t\mathbf{M}_{t-1} - \gamma_t \, u_t \, k_t^\top \in [0, 1] \qquad (8)$$

where, $u_t^\top := \mathbf{J}_\phi \left( \phi(\mathbf{M}_{t-1} \, k_t) - \boldsymbol{\alpha}_t \right)$. This has been recently explored in the context of test-time memorization (Behrouz et al., 2025b; Karami & Mirrokni, 2025; Wang et al., 2025). In this setting, the scalar $\beta_t = 1 - \lambda_t \in [0, 1]$ acts as a forget gate, controlling the retention of prior memory. When $\beta_t \to 1$, it selectively updates the memory based on the interaction of state and input token without fading its magnitude, while $\beta_t \to 0$ erases the memory (possibly due to the change of context). Such scalar gating mechanisms have gained renewed attention in recent RNN architectures as they provide a lightweight yet effective memory update (Beck et al., 2024; Behrouz et al., 2024; Peng et al., 2021b; Sun et al., 2024b; Yang et al., 2024a).

### 3.2 Parallel and Hardware Efficient Implementation

The non-linear nature of the update rule (8) typically hinders straightforward parallelization. Several techniques have been proposed to address this limitation (Gonzalez et al., 2024; Lim et al., 2023). Sun et al. (2024a) introduces mini-batch gradient descent, where the sequence is divided into chunks, and the state at the beginning of each chunk is used to compute the gradients for all time steps within that chunk. In this approach, the gradients within each chunk are approximated as: $\nabla_{\mathbf{M}}\mathcal{L}_t \approx \mathbf{G}_t(\mathbf{M}_{t'}, \boldsymbol{\alpha}_t, v_t)$, where $\mathbf{M}_{t'}$ represents the state at the beginning of the chunk (i.e., the final state from the preceding chunk and $t' = t - \mathrm{mod}(t, C)$ with $C$ denoting the chunk size.

This strategy enables the parallel computation of a mini-batch of *stale* gradients at the start of each chunk, thereby significantly enhancing scalability. Using this approximation effectively linearizes the general non-linear recurrence (equation 8) within each chunk, leading to the following recurrent update rule for the internal state and the memory readout of the compression layer:

$$\{y_t\}_{t=1}^T = \mathrm{compress}(\{q_t, k_t, \boldsymbol{\alpha}_t\}_{t=1}^T) = \begin{cases} \mathbf{M}_t = \mathbf{M}_{t-1} - 2\gamma_t \left( \mathbf{J}_\phi \left( \phi(\mathbf{M}_{t'} \, k_t) - \boldsymbol{\alpha}_t \right) \right) k_t^\top \\ y_t = \mathbf{M}_t q_t \end{cases} \qquad (9)$$

This locally linear recurrence satisfies the associative property and can therefore be parallelized by parallel scan (a.k.a. prefix sum) (Blelloch, 1990), or formulated into a parallel chunkwise form which has been shown to efficiently utilize the `matmul` units of the modern GPUs and can be more I/O efficient (Hua et al., 2022; Kacham et al., 2024; Yang et al., 2024b). We adopt this approach and derive a hardware-optimized chuck-wise form for the recurrence update with state decay in equation 7. For $b$-th block, covering time steps $t$ where $bC + 1 \leq t < (b+1)C$, for $r$-th step in the block, let's define the local cumulative product of decay factors as $\mu_r^b = \prod_{i=bC+1}^{bC+r} \beta_i$. We also denote the segmented cumulative product over the sub-block steps $i$ to $j$ ($1 \leq i \leq j \leq C$) as $\omega_{j,i}^b = \frac{\mu_j^b}{\mu_i^b}$ and the lower triangular matrix with entries $[\boldsymbol{\Omega}^b]_{j,i} = \omega_{j,i}^b \, \forall \, i \leq j$ (and 0 otherwise). By unrolling the recurrence relation locally within the $b$-th block we obtain:

$$\mathbf{M}_r^b = \beta_r^b \, \mathbf{M}_{r-1}^b - \gamma_r^b \, u_r^b \, k_r^{b\top} = \mu_j^b \mathbf{M}^{b-1} - \sum_{i=1}^r \gamma_i^b \, \boldsymbol{\Omega}_{r,i} \, u_i^b \, k_i^{b\top} \qquad (10)$$

Where $\mathbf{M}^{b-1}$ is the state at the end of the $(b-1)$-th-block. Then $\mathbf{M}^b$ can be expressed in matrix form as:

$$\mathbf{M}^b = \mu_j^C \mathbf{M}^{b-1} - \mathbf{U}^b \, \mathrm{Diag}(\gamma \odot \boldsymbol{\Omega}_{B:}) \, \mathbf{K}^{b\top} \qquad (11)$$

where, $\odot$ is the element-wise product. Also, unrolling the readout step and formulating it in matrix form for the entire block yields:

$$\mathbf{Y}^b = \mathrm{Diag}(\boldsymbol{\mu}^b) \, \mathbf{M}^b \mathbf{Q}^b - \mathbf{U}^b \left( \mathbf{K}^{b\top} \mathbf{Q}^b \odot \mathrm{Diag}(\gamma) \odot \boldsymbol{\Omega} \right) \qquad (12)$$

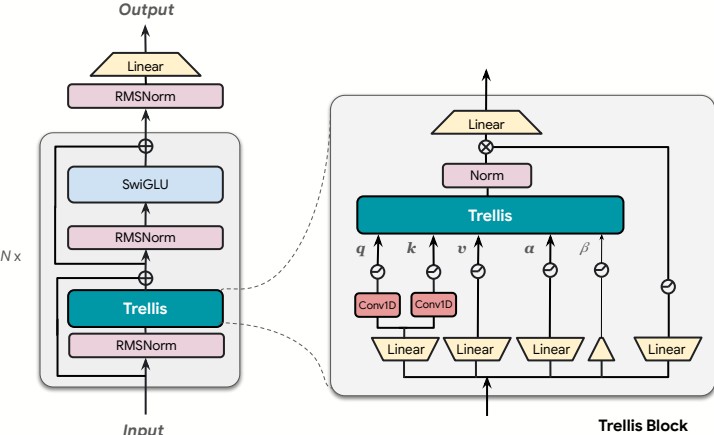

Figure 1: *(Left)* Block diagram of the language model. *(Right)* The Trellis block. Each sequence mixing block is composed of a short Conv1D for $\{q, k\}$ and the Trellis is followed by a post normalization and the GeLU post-gate.

where $\boldsymbol{\mu}^b = [\mu_1^b, \ldots, \mu_C^b]$, and $\mathbf{K}^b, \mathbf{Q}^b \in \mathbb{R}^{d \times C}$ and $\mathbf{U}^b, \mathbf{Y}^b \in \mathbb{R}^{m \times C}$ are matrices collecting the corresponding vectors within the block. This matrix form, which computes the states only at the end of each block, extends the State Space Duality formulation introduced in Mamba2 (Dao & Gu, 2024) and enables efficient use of the `matmul` operations on GPUs.

Consequently, applying the proposed recurrent compression model to key and value caches in a two-pass process yields the following operations:

$$\{\hat{\boldsymbol{y}}_t\}_{t=1}^T = \text{compress}(\{\boldsymbol{q}_t, \boldsymbol{k}_t, \boldsymbol{\alpha}_t\}_{t=1}^T), \quad \{\hat{\boldsymbol{y}}_t, \boldsymbol{\alpha}_t,\} \in \mathbb{R}^m \tag{13}$$

$$\{\boldsymbol{y}_t\}_{t=1}^T = \text{compress}(\{f(\hat{\boldsymbol{y}}_t), \boldsymbol{v}_t, \boldsymbol{\alpha}_t\}_{t=1}^T), \quad \boldsymbol{y}_t \in \mathbb{R}^d, \tag{14}$$

We explored alternative choices for the intermediate activation function in our architecture and found that *normalized SiLU* defined as $f(\boldsymbol{x}) = \frac{\text{SiLU}(\boldsymbol{x})}{\|\text{SiLU}(\boldsymbol{x})\|}$ outperforms the commonly used Softmax, also known as the normalized exponential function. This improvement can be explained by the fact that, in our architecture—where the cache is densely compressed into a limited number of memory slots ($m \ll T$)—a normalization function with *less spikiness than* Softmax is more effective for retrieval. It is also worth noting the difference in input ordering between the second pass of our architecture and that of ABC and GSA. In Trellis, $\boldsymbol{\alpha}_t$ acts as a shared target vector for both compression layers. Consequently, we use $\boldsymbol{y}_t = \phi(\mathbf{M}_t^\top \boldsymbol{q}_t)$ as the readout operation of the value compression layer to ensure correct output dimensions. As a result, in the first pass (equation 13), Trellis compresses the new key embedding, $\boldsymbol{k}_t$, into its memory, generating an intermediate representation and in the second pass (equation 14), it compress the new value embedding and finally outputs $\boldsymbol{y}_t$.

The overall Trellis architecture used for language models is illustrated in Figure 1.

### 3.3 Related Works and Discussion

**Linear State Space Models (SSMs) and Recurrent Neural Networks (RNNs)** have recently received renewed interest as an efficient paradigm for sequence modeling. Offering sub-quadratic scaling during training and constant-time recurrence at inference, they have proven particularly effective for modeling long-range dependencies (Gu et al., 2020). The recurrence in the linear time invariant SSMs can be reformulated as a global convolution, enabling efficient computational implementations (Gu et al., 2021; Mehta et al., 2022). Furthermore, the linearity in input-dependent SSMs (Dao & Gu, 2024; Gu & Dao, 2023) and

modern gated RNNs (Beck et al., 2024; De et al., 2024; Orvieto et al., 2023) take advantage of parallelization through techniques like associative scan (Blelloch, 1990; De et al., 2024; Smith et al., 2023), or chunkwise parallel forms (Behrouz et al., 2024; 2025a; Hua et al., 2022; Yang et al., 2024b;c). However, a potential limitation arises from state-independent updates. The additive (gated) linear modification of the memory, doesn't consider the interaction between the memory content and incoming keys, potentially limiting the model's ability to efficiently compress and summarize information within its finite memory state.

**Fast Weight Programmers (FWPs)** represent a class of architectures where the parameters of one network, often termed the "fast" network, are dynamically generated or modified by another "slow" network (Schlag et al., 2021; Schmidhuber, 1992). This concept, also referred to as input-dependent parameterization (Gu & Dao, 2023; Karami & Ghodsi, 2024; Karami et al., 2019), enables the model to adapt to the specific characteristics of its input, potentially capturing more complex contextual patterns. In our framework, the compression layer's memory state acts as "fast weights" that quickly learn in-context information, while the "slow" network parameters learns generalizable patterns across the training set. Therefore, we effectively adopt a meta-learning (Andrychowicz et al., 2016; Bengio et al., 1990; Schmidhuber, 1992; Thrun & Pratt, 1998) style approach by deriving an internal online learning procedure that continuously minimizes a reconstruction loss.

**Online Gradient Descent** is widely adopted for non-stationary sequential data in classical adaptive filtering such as the Least Mean Squares (LMS) algorithm (Haykin, 2002). LMS-also known as the *Delta Rule* (Schlag et al., 2021; Widrow & Hoff, 1988)-typically minimizes the instantaneous squared error through a simple a linear update rule. Convex optimization-based methods have also been explored for identifying SSMs (a.k.a. linear dynamical systems); for example, Karami et al. (2017) formulates the problem as a multi-view matrix factorization and proposes a global optimizer.

In contrast, our work offers a more general non-linear update rule incorporating a forgetting gate. This update rule is applied to both key and value compression within a cascading, two-pass structure. This design is particularly suitable for long-context modeling, where efficient and expressive memory compression is critical.

## 4    Experiments

**Overview:**    In this section, following recent studies (Lim et al., 2023; Zhang et al., 2024), we evaluate the performance of Trellis in various downstream tasks, including language modeling, common-sense reasoning, needle in haystack, and time series forecasting. We use four scales of the proposed Trellis with (1) 125M, (2) 350M, (3) 780M, and (4) 1B parameters. For each scale, we train three different versions, each of which is solely trained on either Pile (Gao et al., 2020), C4 (Raffel et al., 2020), or Books datasets (a subset of the Pile). The main reason for our choice of datasets and training different version of Trellis are two folds: First, the dataset used for training and its characteristics can significantly affect the performance of the model. Therefore, while a dataset can be more useful for a model, it might not be the best choice for others. Accordingly, to avoid cherry-picking of the dataset, and to emphasize the generalizebility and the power of Trellis, we use three different versions, each of which is trained on a commonly-used dataset in the literature. Second, we use each dataset to highlight the strength of Trellis in one aspects. That is, we use Pile (resp. Books) dataset to highlight the strength of Trellis in short context (resp. long context).

**Baseline Models:**    We compare our method against the Transformer++ architecture proposed in Touvron et al. (2023) and the following sub-quadratic models: Linear-Attention (LA) (Katharopoulos et al., 2020), TTT (Sun et al., 2024a), GSA (Zhang et al., 2024), DeltaNet (Yang et al., 2024c), Gated-DeltaNet (Yang et al., 2024a), Mamba2 Dao & Gu (2024). All RNN models follow the Mamba architecture where the sequence models follows by a normalization and gating before output linear projection.

To investigate the effect of context length on model performance, in this part, we evaluated models trained with various context lengths, comparing Trellis (both with and without its forget gate) against the baseline models. For the Books dataset, models were trained

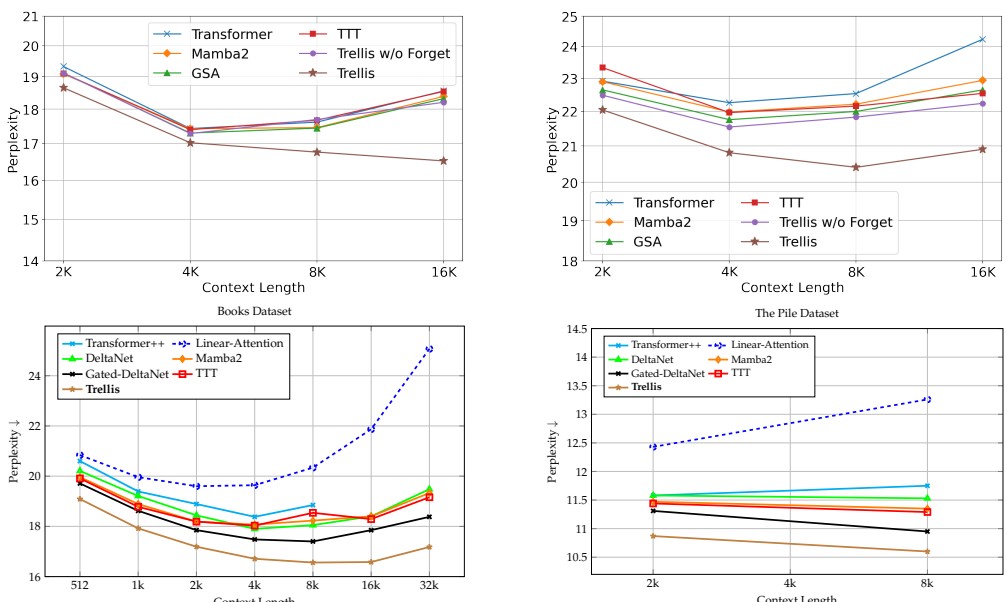

Figure 2: The effect of context length on model's perplexity. Subplots show: *(Top Left)* C4 dataset with 350M parameters; *(Top Right)* C4 dataset with 780M parameters; *(Bottom Left)* Books dataset with 125M parameters; *(Bottom Right)* The Pile dataset with 125M parameters. Training Transformers from scratch on very long sequence contexts (e.g., $T \in \{16k, 32k\}$) can yield poor perplexity, hence the standard practice for such contexts is typically to finetune a Transformer pre-trained on shorter sequences (Touvron et al., 2023). Here, for the Transformer baseline trained from scratch in these results, performance is only reported up to a context length of $T = 8k$.

Table 1: Performance of Trellis and baselines on language modeling and common-sense reasoning tasks.

| Model | LMB. ppl↓ | LMB. acc↑ | PIQA acc↑ | Hella. acc↑ | Wino. acc↑ | ARC-e acc↑ | ARC-c acc↑ | SIQA acc↑ | Avg. ↑ |
|---|---|---|---|---|---|---|---|---|---|
| 790M params / 30B tokens | | | | | | | | | |
| Transformer++ | 25.89 | 33.41 | 64.75 | 41.98 | 51.33 | 59.06 | 31.85 | 40.26 | 46.09 |
| Mamba2 | 28.91 | 32.72 | 64.98 | 42.60 | 50.01 | 61.99 | 30.24 | 41.07 | 46.23 |
| TTT | 27.05 | 33.18 | 65.03 | 43.17 | 49.93 | 62.16 | 32.13 | 41.35 | 46.71 |
| Gated-DeltaNet | 21.40 | 34.83 | 65.79 | 43.66 | 50.45 | 64.02 | 32.24 | 41.68 | 47.52 |
| Trellis | 20.28 | 35.44 | 67.51 | 44.29 | 51.08 | 65.12 | 33.17 | 42.04 | 48.38 |

using sequence lengths of $\{512, 1024, 2048, 4096, 8192, 16384, 32748\}$. We also trained the models on the C4 dataset with context lengths in $\{2048, 4096, 8192, 16384\}$, and on the Pile dataset using a subset of context lengths $T \in \{2048, 8192\}$. The results presented in Figure 2 show that Trellis achieves the lowest perplexity compared to all baselines across all tested context lengths. Interestingly, Trellis shows greater performance gains compared to other linear RNNs as the sequence length increases, demonstrating the potential of our approach for tasks requiring long-context reasoning. Furthermore, comparing against Trellis w/o forgetting highlights the importance of this component in the overall performance of Trellis.

**Language Modeling and Common-sense Reasoning** Following recent studies on sequence modeling (Behrouz et al., 2024; Yang et al., 2024a), in this section, we compare the performance of Trellis with modern linear recurrent neural networks and Transformer on language modeling and common-sense reasoning tasks. The results are reported in Table 1. Trellis achieves outstanding performance across all scales and outperform all linear recurrent models and Transformer++.

Table 2: Ablations on improving from linear DeltaNet (Yang et al., 2024c) and also TTT (Sun et al., 2024a). All models have 125M parameters trained on The Pile dataset.

| Configuration | ppl ↓ |
|---|---|
| DeltaNet | 11.58 |
| TTT | 11.44 |
| Trellis | 10.87 |
|   w/o forget gate | 11.28 |
|   with f = L2-SiLU | 10.98 |
|   with f = Softmax | 11.29 |
|   Linear f = Softmax | 12.71 |
|   Linear f = LN-SiLU | 11.65 |
|   m = 32 | 11.14 |
|   m = 128 | 10.87 |
|   b = 1 | 10.75 |

Table 3: Performance of Trellis and baselines with 1B parameters on S-NIAH task from RULER benchmark. The best results with highest accuracy are highlighted.

| Model | S-NIAH-PK | | | S-NIAH-N | | | S-NIAH-W | | | Average |
|---|---|---|---|---|---|---|---|---|---|---|
| | 2K | 4K | 8K | 2K | 4K | 8K | 1K | 2K | 4K | |
| TTT | 98.4 | 98.8 | 98.0 | 60.2 | 36.6 | 10.2 | 85.8 | 78.8 | 28.0 | 66.1 |
| Mamba2 | 98.6 | 61.4 | 31.0 | 98.4 | 55.8 | 14.2 | 62.2 | 42.2 | 4.2 | 52.0 |
| DeltaNet | 96.8 | 98.8 | 98.6 | 47.2 | 15.4 | 12.8 | 85.2 | 46.2 | 20.0 | 57.9 |
| Gated DeltaNet | 89.8 | 91.4 | 90.0 | 99.2 | 91.8 | 26.4 | 86.4 | 82.6 | 24.4 | 75.8 |
| Trellis | 99.2 | 95.2 | 97.8 | 99.4 | 94.2 | 34.6 | 86.4 | 82.8 | 28.4 | 79.8 |

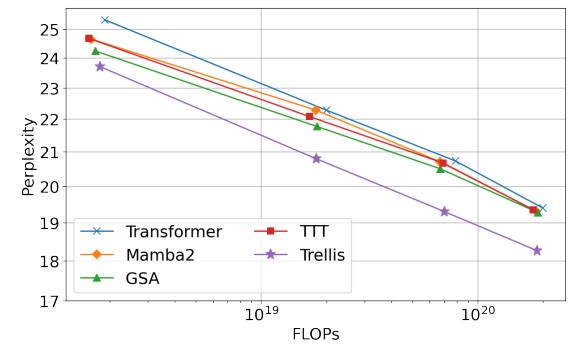

Figure 3: Scaling pattern of models w.r.t. Perplexity vs. FLOPs.

To see the scaling law in Trellis and compare it with baselines, we report the perplexity for different model sizes in Figure 3. Trellis shows a consistent trend and achieves better perplexity compared to baselines with a fixed budget of FLOPs. This pattern shows that in the trade-off of efficiency and effectiveness, Trellis achieves Pareto frontier results.

**Needle in Haystack Tasks** In this section, we follow recent studies on sequence modeling and evaluate the performance of Trellis in the RULER benchmark and needle in haystack tasks (Hsieh et al., 2024). The results are reported in Figure 3. Trellis outperforms all the baselines with +4% performance gain on average over the second best model, showing higher performance gain of about +6% in longer sequences. We attribute the superior performance of Trellis to its: (1) more powerful memory management using the hybrid of linear and non-linear recurrence with forget gate, (2) architectural design, in which, we use a two-pass recurrence with *the same memory* to store keys and values. In the results we find two exceptions that Trellis achieves the second best result (i.e., S-NIAH-PK task with 4K and 8K sequence length). These results matches the observation of Yang et al. (2024a) that simple NIAH task with repeated synthetic context require long-term retention, which a forget gate can damage.

**Ablations** In this section, we perform ablation studies on key components and design choices within the proposed architecture to evaluate the contribution of each to the overall performance. The baseline Trellis is a 2-pass K-V cache compression layer (presented in equations (13) and (14)) with 125M parameter ($d = 768$ and $m = 64$). Its intermediate activation function: $f(x) = \text{LN-SiLU}(x) := \text{LayerNorm}(\text{SiLU}(x))$. Starting from this baseline configuration, we evaluate the impact of the following modifications (one component at a time): (1) Removing forget gate, (2, 3) Changing the intermediate activation function: normalized SiLU: $f(x) = \frac{\text{SiLU}(x)}{\|\text{SiLU}(x)\|}$ and standard softmax, (4, 5) Removing the compression layer non-linearity, *i.e.* $\phi(x) = x$ in equation 4, which reduces the recurrence in both passes to the Delta Rule. This linear recurrence is then tested with $f = \text{Softmax}$ and $f = \text{LN-SiLU}$. (6, 7) Varying the memory size, and (8) Using fully non-linear recurrence with mini-batch (chunk) size: $B = 1$. The results, summarized in Table 2, highlight the significance of all

these design choices on overall performance, notably, the 2-pass compression, forget gate and the choice of intermediate gate.

Table 4: Performance comparison of 125M-parameter language models on The Pile (2k context length) and Books (various context lengths) datasets. Baselines include Transformer++ (Touvron et al., 2023), Linear-Attention (LA) (Katharopoulos et al., 2020), DeltaNet (Yang et al., 2024c), Gated-DeltaNet (Yang et al., 2024a), Mamba2 (Dao & Gu, 2024), and TTT (Sun et al., 2024a). The best results are highlighted.

| Model | Pile (2k) | Pile (8k) | Books (512) | Books (1k) | Books (2k) | Books (4k) | Books (8k) | Books (16k) | Books (32k) | Pile (2k) |
|---|---|---|---|---|---|---|---|---|---|---|
| | ppl↓ | ppl↓ | ppl↓ | ppl↓ | ppl↓ | ppl↓ | ppl↓ | ppl↓ | ppl↓ | ppl↓ |
| | *125M params / 2.4B tokens* | | | | | | | | | *350M params / 7.5B tokens* |
| Transformer++ | 11.58 | 11.75 | 20.60 | 19.39 | 18.89 | 18.38 | 18.85 | 29.41 | | 8.48 |
| Linear-Attention | 12.43 | 13.26 | 20.84 | 19.96 | 19.60 | 19.64 | 20.34 | 21.87 | 25.07 | 9.16 |
| DeltaNet | 11.58 | 11.53 | 20.21 | 19.21 | 18.44 | 17.90 | 18.05 | 18.40 | 19.48 | 8.62 |
| Mamba2 | 11.47 | 11.35 | 19.96 | 18.89 | 18.19 | 18.07 | 18.23 | 18.40 | 19.33 | 8.56 |
| Gated-DeltaNet | 11.31 | 10.95 | 19.71 | 18.62 | 17.85 | 17.48 | 17.40 | 17.85 | 18.38 | 8.53 |
| TTT | 11.44 | 11.29 | 19.91 | 18.78 | 18.19 | 18.03 | 18.54 | 18.29 | 19.16 | 8.62 |
| Trellis | 10.87 | 10.60 | 19.09 | 17.92 | 17.19 | 16.71 | 16.56 | 16.58 | 17.18 | 8.26 |

## 5  Conclusion

In this paper, we introduced Trellis, a meta in-context learning framework that learns how to compress the KV cache of attention into a fixed-size memory. Trellis employs a two-pass recurrent memory update process, in which keys and values are stored in a compact memory module. To enhance learning from long context, Trellis further incorporates a gating mechanism that learns how to filter and forget irrelevant past information. Our experimental results across diverse domains—including language modeling, commonsense reasoning, needle-in-a-haystack tasks, and time series data—indicate the superior performance of Trellis compared to modern linear RNNs and Transformers. The results further support the new architectural design of Trellis, showing that all its components meaningfully contribute to its overall performance. By introducing learned, dynamic memory compression, Trellis offers an effective and efficient solution for long-sequence modeling.

Looking ahead, recent advances in non-linear recurrent models such as Titans (Behrouz et al., 2024; 2025a), Lattice (Karami & Mirrokni, 2025) offer promising directions for further improving KV cache compression. Additionally, finetuning pretrained Transformers into RNNs (T2R) (Kasai et al., 2021) represents another promising direction to explore the capabilities of large pre-trained models with the inference efficiency of recurrent architectures.

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

# A  Experiment Details

**Architectural Details.** We provide the architectural details of our model in Table 5.

Table 5: Architectural Details.

| Model | Block | Dim | Head | Peak LR | Token |
|-------|-------|------|------|---------|-------|
| 125M | 12 | 768 | 12 | 3e-3 | 2.4B |
| 170M | 12 | 768 | 16 | 3e-3 | 15B |
| 350M | 24 | 1024 | 16 | 1.5e-3 | 15B |
| 780M | 24 | 1536 | 16 | 1.25e-3 | 30B |

**The Choice of Datasets and Training.** We train our models on three different datasets. Our goal is to highlight the generalizability and power of Trellis across diverse datasets. For smaller and medium-sized models (125M and 350M parameters), we included a broader set of baselines and conducted detailed ablation studies. Due to computational constraints, we limited the training of larger models (780M and 1B parameters) to comparisons with the most relevant baselines. Note that Tables 1 and 3 present results for models with 780M and 1B parameters, respectively, to show Trellis's performance at larger scales. The results in the bottom row of Figure 2 are based on models trained on The Pile and Books datasets. All ablation studies and Table 4 are also conducted using models trained on The Pile. The models used in the remaining experiments (including Table 3 and Figure 3) are trained on the C4 dataset.

