# OpenReview forum: "TRELLIS: Learning to Compress Key-Value Memory in Attention Models"
_colmweb.org/COLM/2025/Conference — COLM 2025_

### Official Review · Reviewer_M9Vs · 2025-05-09

**Rating:** 6
**Confidence:** 3
**Ethics Flag:** 1

**Summary:**

In this paper, the authors propose Trellis: a new method that allows transformers to have a bounded memory (compressed KV cache) which is updated at each step and contains a forget gate that allows the model to selectively forget past information.

This work is inspired by ABC (Attention-with-Bounded-Memory Control) and GSA (Gated Slot Attention). But differently from ABC and GSA, the authors approach the memory’s update as an online regression problem, updating the memory using a gradient descent step per token.
Similarly to GSA, the memory update includes a forget gate which allows the model to forget past information. This is accomplished by applying an l_2 regularization in the loss.

To ensure its applicability the authors provide a parallel and efficient implementation of Trellis.

The authors compare the approach with baselines comprising recurrent and efficient transformer architectures on language modelling, common-sense reasoning tasks, needle in the haystack, and time series.
These comparisons show that TRELLIS outperforms the baselines in most tested tasks across datasets and model parameters. Moreover, the authors show that for the same number of FLOPs Trellis leads to a lower perplexity than Mamba2, TTT, and GSA.

Overall, I believe the proposed approach is interesting and applicable to a wide range of tasks. However, the experimental section is quite disappointing.

**Questions To Authors:**

-	In lines 96 to 102, you mention “… the additive update rule in ABC and GSA endlessly accumulates new tokens into the fixed memory space without an explicit mechanism to deallocate past information …”.
However, the GSA contains a gating mechanism used to forget past information. Can you explain why you make this statement?
-	See “reasons to reject”.

**Reasons To Accept:**

-	According to the experiments, the approach proposed seems to outperform other efficient architectures.
-	The authors show that Trellis’ implementation can be efficient and parallelizable.

**Reasons To Reject:**

-	The experimental section is quite disappointing and missing details.
First, I did not find an explanation as to why different baselines were used for different experiments (for example in figure 2 the two top plots have a list of baselines completely different than the that of the two bottom plots, and figures 4 or 5 even have a smaller number of baselines).
Then, the authors mention that they train models of 4 model sizes up to 1B parameters on the 3 datasets. However, most of the comparisons show results for models with only 125M parameters or 780M parameters. Why is this?
-	Nowadays, we see LLMs with context lengths bigger than 100k tokens. However, the paper experiments are limited to context lengths of 16k or 32k. I understand that experiments with bigger models trained with more tokens and with bigger context lengths are costly. But this limitation makes it difficult to understand the model’s applicability to real-world scenarios.

---

> ### Author Response · Authors · 2025-06-03
>
> We thank the reviewer for their constructive feedback and valuable observations. In the following, we address their concerns and respond to questions.
>
> > I did not find an explanation as to why different baselines were used for different experiments
>
> Our goal in training models on three different datasets, reported in Figure 2 (and also Table 4) was to highlight the generalizability and power of Trellis across diverse datasets. For smaller and medium-sized models (125M and 350M parameters), we included a broader set of baselines and conducted detailed ablation studies. Due to computational constraints, we limited the training of larger models (780M and 1B parameters) to comparisons with the most relevant baselines. We note that Table 1 and Figure 4 present results for models with 780M and 1B parameters, respectively, to show Trellis’s performance at larger scales. We acknowledge that the variation in baseline coverage across plots requires a clarification, and we will make these choices explicit in the experimental details section of the final version.
>
> > Nowadays, we see LLMs with context lengths bigger than 100k tokens.
>
> We acknowledge the reviewer’s concern regarding context length greater than 100k tokens. In this work, we limited our evaluation to a maximum of 32K tokens, in line with the settings used in the main baseline papers. Moreover, widely used long-context evaluation suites for language modeling, such as LongBench [1], include tasks with sequences that rarely exceed 32K tokens with the average context lengths in these benchmarks typically range from ~1K to maximum of ~22K.
>
> > In lines 96 to 102, you mention “… the additive update rule in ABC and GSA endlessly accumulates new tokens into the fixed memory space without an explicit mechanism to deallocate past information …”. However, the GSA contains a gating mechanism used to forget past information.
>
> While it is true that GSA incorporates a gating mechanism, it operates within an additive (Hebbian-like) update form, where forgetting is implemented as a *decay (fading) of the entire memory content.* In contrast, the update rule in Trellis, derived from online optimization (Eqs. 5–6), is a form of Fast Weight Programmers (FWPs) and generalizes the classical Delta Rule (Widrow & Hoff, 1988; Schlag et al., 2021) (we can verify it by simply removing the nonlinearity (i.e., setting $\phi(\cdot)$ to the identity function), our update reduces to the Delta Rule). The Delta Rule recurrence can be written as (Schlag et al., 2021, Yang et al., 2024b):
>
> $$ \mathbf{S}_t = \mathbf{S}\_{ t - 1} - \underbrace{\mathbf{v}_t^{\text{old}}  \mathbf{k}_t^\top}\_{remove} + \underbrace{\mathbf{v}_t^{\text{new}}  \mathbf{k}_t^\top}\_{write},
> \quad \text{where }  \mathbf{v}_t^{\text{old}}  = \mathbf{S}\_{t-1} \mathbf{k}_t , \quad  \mathbf{v}_t^{\text{new}} = \beta_t \mathbf{v}_t + (1 - \beta_t)\mathbf{v}_t^{\text{old}}
> $$
>
> Therefore, unlike gated additive updates in GSA, which modulates how fast past memory fades, Delta Rule provides a mechanism to *deallocate or overwrite irrelevant past information* based on the interaction between incoming keys and the current state, offering higher memory capacity and greater flexibility. The non-linear generalization used in the two-pass process in Trellis is demonstrated in the experiments that yields stronger empirical performance.
>
>
> References:
>
> [1] Bai, Yushi, et al. "Longbench: A bilingual, multitask benchmark for long context understanding." arXiv preprint arXiv:2308.14508 (2023).

---

> > ### Comment · Reviewer_M9Vs · 2025-06-05
> >
> > Having read the authors comment, I do understand the authors reasoning and thus I will slightly increase my rating. However some of my concerns are still present.
> >
> > - I do understand that it is costly to train so many baselines on several datasets for 4 different sizes.
> > However, I still don't understand the reasoning behind the choice of number of parameters and baselines for each experiment and think that this needs to be clearly stated in the paper.
> > For example, what's the reasoning behind changing the set of baselines from table 1 to figure 4 (replacing Transformer ++ with DeltaNet) and changing the model sizes (I believe the authors have a perfectly good explanation for this, but it isn't stated on the paper)? And the same for figure 5? Why do the relevant baselines change from experiment to experiment?
> > I also struggle to understand with which datasets were trained the models used in each experiment.
> >
> > - I understand the difference between the two gating mechanisms. However, I still think the comment is not correct and that it should be edited.

---

> > > ### Author Response · Authors · 2025-06-07
> > >
> > > We thank the reviewer for their follow-up and for reconsidering their rating. We will add detailed clarification in the final version regarding the experimental setup, including the datasets used and the reasoning behind the choice of model sizes and baselines for each experiment. Specifically: The results in the bottom row of Figure 2 are based on models trained on The Pile and Books datasets. All ablation studies and Table 4 are also conducted using models trained on The Pile. The models used in the remaining experiments (including Figures 4 and 5) are trained on the C4 dataset.
> > >
> > > Regarding the distinction between the DeltaNet update rule and Hebbian-style updates (used in GLA): we will revise the text to more clearly describe this difference.

---

### Official Review · Reviewer_1Hgq · 2025-05-13

**Rating:** 10
**Confidence:** 4
**Ethics Flag:** 1

**Summary:**

The Trellis paper presents a novel and ambitious Transformer design to tackle the critical issue of the growing Key-Value cache. It enables the model to learn dynamic compression of this memory into a fixed-size store. This core concept of a learnable, adaptive KV cache, managed through an online gradient descent procedure and a forget gate, is a highly original and principled approach for efficient long-context modeling. The methodology is clearly explained, reflecting a solid grasp of Transformer mechanics, optimization, and hardware efficiency considerations. The work is commendably supported by extensive experiments across diverse tasks.  To further strengthen its claims and offer an even more comprehensive view of its standing, the experimental validation would benefit from including direct comparisons with other recent and highly relevant architectures designed for efficient long-context processing, such as Titans and RWKV. Benchmarking against these specific models would provide a more complete understanding of Trellis's advantages within this rapidly evolving research landscape. Despite this point for further investigation, Trellis presents a significant and innovative contribution. It offers a compelling and well-engineered solution to a crucial problem in scaling Transformers, supported by robust empirical evidence of its effectiveness and efficiency. This makes it a high-quality paper with substantial potential to influence future work in efficient sequence modeling.

**Reasons To Accept:**

mentioned in summary

**Reasons To Reject:**

N/A

---

> ### Author Response · Authors · 2025-06-02
>
> We sincerely thank the reviewer for their insightful and encouraging feedback. We are especially grateful for their recognition of the novelty, and potential impact of our work and appreciate the acknowledgement of both the technical depth of our approach and the breadth of our experimental validation.

---

### Official Review · Reviewer_RaMN · 2025-05-13

**Rating:** 6
**Confidence:** 4
**Ethics Flag:** 1

**Summary:**

This paper introduces Trellis, a novel Transformer variant designed to address the scalability and memory bottlenecks of attention mechanisms in long-context sequence modeling. Trellis replaces the traditional key-value (KV) cache with a learned, bounded memory using a two-pass compression mechanism. The memory is updated using an online gradient descent procedure with a forget gate, allowing it to dynamically compress incoming key-value pairs while maintaining relevant context. The method is evaluated on a variety of tasks—language modeling, reasoning, and time-series forecasting—where it consistently outperforms strong baselines, particularly at longer sequence lengths.

**Reasons To Accept:**

* The paper addressed the fundamental limitation in Transformer attention. It proposed several techniques on top of previous SSM to improve the performance.

* It has shown strong performance against previous Transformer variants.

**Reasons To Reject:**

* The paper has limited novelty. The main structure is following previous SSM papers like Mamba. It has proposed some variants on top of it. However, the paper has not made clear the difference between different architectures.

* The paper has only conducted Single Needle in A Haystack evaluation. For SSM models with limited memory size, memorization capability is important and should be measured in detail. The forget and state decay mechanism may be able to retrain a short info but cannot hold the entire previous information history. It is important to study the limitation in SSM-type of models to make a fair comparison against Transformer.

* The current setting follows the previous pattern (1) propose a new architecture (2) train with limited number of tokens and controlled number of parameters. (3) evaluate on a set of general language modeling benchmarks.
It would be really interesting to jump out of this box and see experiments around more fundamental capabilities around Transformers and SSMs.

---

> ### Author Response · Authors · 2025-06-03
>
> We thank the reviewer for their feedback and for highlighting the strong performance of our work.  In the following, we address their concerns and, in particular, further clarify the key novelties of our proposed method.
>
> > *The paper has limited novelty. The main structure is following previous SSM papers like Mamba.*
>
> We highlight several fundamental differences and novelties in our approach, which are key to its strong performance and distinguish it from the available SSM architectures like Mamba.
> - Trellis is derived from the principles of online optimization for representation learning, leveraging techniques from meta-learning and test-time training. Its update rule is a form of online gradient descent on reconstruction losses to update the internal memory of the pairs of Keys/Values in the transformers.
> - It learns how to compress Key-Value pairs into a bounded memory via a **non-linear, two-pass recurrent process** which also takes advantage of parallel and hardware efficient implementation.
> - We tailored a **different form of two-pass process** in equations (13) (14) compared to prior work like ABC and incorporates a normalized SiLU activation, which our ablation studies demonstrate is highly effective for retrieval in bounded memory.
> - We also derived a state decay mechanism based on $l_2$ regularization in the inner loop. The overall update rule is **state and input--dependent** (the update for a memory slot depends on the current value of that slot and also the layer’s input).
>
> In the above we highlighted the key difference of the proposed method vs the selective SSM architectures like Mamba. While Trellis architecture fundamentally takes advantage of **non-linear update rules which is state-dependent and input dependent in a novel two-pass process**, Mamba’s gating mechanism is input dependent and updates a single associative memory in a linear form.  Furthermore, the derivation of our model from an online optimization perspective is different theoretically from Mamba's derivation from continuous-time systems.
>
> >  The forget and state decay mechanism may be able to retrain a short info but cannot hold the entire previous information history.
>
> As discussed in section 2, while Transformer-based models benefit from an unbounded KV cache, the SSM and RNN alternatives have bounded state size (memory), hence their ability to recall the previous information history is inherently limited and they can’t memorize the entire context in long sequence. Given this limitation,  the objective of this work is not to achieve lossless recall, but to design a non-linear recurrence that *efficiently* compress new keys and values into the memory while minimizing information loss. We highlight two key aspects of the proposed design for effective long-term memory management:
>
> 1. **Adaptive State Decay:** Regarding the concern on state decay (forget), the forget gate in Trellis is not a simple, constant decay of past information (as seen in models like S4 or the weight decay in deep neural network training). Instead, it is an *adaptive input-dependent gating mechanism* that learns to adjust its decay based on the current input, enabling it to selectively preserve important past information while making room for new context. Notably, we highlight that the forget gate weights are biased towards one by design, ensuring that the model's default behavior is to retain information unless the input dictates otherwise.
> 2. **Nonlinear Memory Update Rule**: Furthermore, the core of Trellis's memory management lies in its update rule, derived from online optimization (Eqs. 5-6). Its non-linear update is a generalization of the Delta Rule  (Widrow & Hoff, 1988; Schlag et al., 2021). By simply removing the nonlinearity (setting $\phi(\cdot)$ to identity in Eqs. 5-6), our update reduces to the Delta Rule, which is known to offer higher memory capacity and greater flexibility compared to the simple additive (Hebbian-like) updates used in most other SSMs, such as Mamba. Importantly, it provides a mechanism to deallocate or overwrite irrelevant past information, which simple additive updates do not (https://openreview.net/forum?id=r61s1FNYlj&noteId=8dqH0mSWc2).
>
> Together with our modified two-pass process, these mechanisms render Trellis an effective long-context sequence model. This is empirically validated in Figure 2, where Trellis's performance advantage over baselines like GSA and DeltaNet becomes more pronounced as the context length increases to 32K.
> Additionally, we evaluated the memorization capability of Trellis in the *RULER benchmark*,  a widely adopted suite specifically designed to evaluate long-context language models. We assess performance on three increasingly complex tasks: *passkey retrieval (S-NIAH-PK), numerical needle in haystack (S-NIAH-N), and word-based needle in haystack (S-NIAH-W)*. Finally, our ablation studies further characterize the effect of higher memory size ($m$) on performance improvement.

---

> > ### Author Response · Authors · 2025-06-03
> >
> > > The current setting follows the previous pattern ….
> >
> > Our primary motivation for adopting the current experimental setting, which is a standard and widely adopted setting in the field, was to ensure a *fair and direct comparison against existing state-of-the-art methods* under similar training budgets. By adhering to this controlled setup, including controlled parameter counts and training data sizes, we can directly measure performance improvements of the proposed model over prior models. That said, we agree that exploring the fundamental capabilities of new architectures beyond standard benchmarks is an exciting direction, and we view this as a promising avenue for future work.
> >
> >
> > References:
> >
> > [1] Gu, Albert, Karan Goel, and Christopher Ré. "Efficiently modeling long sequences with structured state spaces." arXiv preprint arXiv:2111.00396 (2021).

---

> > > ### Comment · Reviewer_RaMN · 2025-06-05
> > >
> > > Thanks for the response. Yeah, I am hoping to see more fundamental research on the capabilities and limitations for the state space model compared to normal Transformer. Besides that, I don't have further questions.

---

### Decision · Program_Chairs · 2025-07-08

**Decision:**

Accept

**Comment:**

This paper introduces a method (Trellis) which compresses the KV cache of transformers, addressing the scalability and memory bottlenecks of attention mechanisms in long-context sequence modeling. The method is inspired by ABC (Attention-with-Bounded-Memory Control) and GSA (Gated Slot Attention) but adds several innovations. The compressed memory is bounded and updated using an online gradient descent procedure with a forget gate. The method is evaluated on language modeling, reasoning, and time-series forecasting with convincing performance for long sequences, outperforming baselines comprising recurrent and efficient transformer architectures, Mamba2, TTT, and GSA.

All reviewers feel positively about this paper, and so do I. While the paper has some drawbacks (somewhat limited novelty and lack of details in the experiments) the proposed approach is interesting, efficiently implemented, and appears to be effective in several tasks. I recommend acceptance, encouraging the authors to address the feedback provided by the reviewers.

[Automatically added comment] At least one review was discounted during the decision process due to quality]